# Computational analysis of GAL pathway pinpoints mechanisms underlying natural variation

**Jiayin Hong**[1,2], **Julius Palme**[1], **Bo Hua**[1], **Michael Springer**[1]*

**1** Department of Systems Biology, Harvard Medical School, Boston, Massachusetts, United States of America, **2** Center for Quantitative Biology and Peking-Tsinghua Center for Life Sciences, Academy for Advanced Interdisciplinary Studies, Peking University, Beijing, China

* michael_springer@hms.harvard.edu

**Data Availability Statement:** The GAL induction profiles, and source code for mathematical modeling and in silico simulations are available at https://github.com/JiayinHong/Natural-variation-in-yeast-GAL-pathway-induction.

## Abstract

Quantitative traits are measurable phenotypes that show continuous variation over a wide phenotypic range. Enormous effort has recently been put into determining the genetic influences on a variety of quantitative traits with mixed success. We identified a quantitative trait in a tractable model system, the GAL pathway in yeast, which controls the uptake and metabolism of the sugar galactose. GAL pathway activation depends both on galactose concentration and on the concentrations of competing, preferred sugars such as glucose. Natural yeast isolates show substantial variation in the behavior of the pathway. All studied yeast strains exhibit bimodal responses relative to external galactose concentration, i.e. a set of galactose concentrations existed at which both GAL-induced and GAL-repressed subpopulations were observed. However, these concentrations differed in different strains. We built a mechanistic model of the GAL pathway and identified parameters that are plausible candidates for capturing the phenotypic features of a set of strains including standard lab strains, natural variants, and mutants. *In silico* perturbation of these parameters identified variation in the intracellular galactose sensor, Gal3p, the negative feedback node within the GAL regulatory network, Gal80p, and the hexose transporters, HXT, as the main sources of the bimodal range variation. We were able to switch the phenotype of individual yeast strains *in silico* by tuning parameters related to these three elements. Determining the basis for these behavioral differences may give insight into how the GAL pathway processes information, and into the evolution of nutrient metabolism preferences in different strains. More generally, our method of identifying the key parameters that explain phenotypic variation in this system should be generally applicable to other quantitative traits.

## Author summary

Microbes adopt elaborate strategies for the preferred uptake and use of nutrients to cope with complex and fluctuating environments. As a result, yeast strains originating from different ecological niches show significant variation in the way they induce genes in the galactose metabolism (GAL) pathway in response to nutrient signals. To identify the

**Funding:** JH was supported by the China Scholarship Council for one-year study at Harvard University (File No. 201606010258). MS is supported by National Institutes of Health under grant no. R01- GM120122-03. The funders had no role in study design, data collection and analysis, decision to publish, or preparation of the manuscript.

**Competing interests:** The authors have declared that no competing interests exist.

mechanistic sources of this variation, we built a mathematical model to simulate the dynamics of the galactose metabolic regulation network, and studied how parameters with different biological implications contributed to the natural variation. We found that variations in the behavior of the galactose sensor Gal3p, the negative feedback node Gal80p, and the hexose transporters HXT were critical elements in the GAL pathway response. Tuning single parameters *in silico* was sufficient to achieve phenotype switching between different yeast strains. Our computational approach should be generally useful to help pinpoint the genetic and molecular bases of natural variation in other systems.

## Introduction

Complex cellular functions are governed by interactions between DNA, RNA, and proteins, forming molecular circuits and regulatory networks. Mathematical models have been useful for gaining insights into the behavior and biological function of such networks. For instance, computational studies have identified specific network topologies that are especially well adapted to performing particular functions such as adaptation [1], oscillation [2–5], cell polarization [6], fold-change detection [7,8], and noise attenuation [9,10]. Mathematical modeling has also broadened our knowledge of how certain fundamental characteristics of biological systems are implemented, especially robustness, which has been studied in the cell cycle [11–13], in bacterial chemotaxis [14], in signaling pathways [15], and in pattern formation and developmental control circuits [16–19]. The new frontier for mathematical modeling is to interpret the network properties that allow individual-to-individual and between-species variation in the behavior of a system while maintaining robust function. Individual-to-individual variation is key to allow populations to "hedge their bets", permitting survival of at least a fraction of the population even under harsh conditions, and between-species variation is required for evolutionary adaptation.

Model organisms are powerful systems in which to study how the genetic architecture of a system supports varied behavior. In particular, the rich ecological history that is readily available for study in yeast species makes yeast an important model of quantitative genetic variation and speciation in evolution. The yeast galactose utilization pathway (GAL pathway) is a well-characterized model system that has made major contributions to our understanding of many features of eukaryotic biology, including the role of noise and stochasticity in eukaryotic gene expression [20,21], and the mechanistic underpinnings of bimodality and bistability in gene expression [22,23]. It is a key model system for the adaptation of microbes to fluctuating and complex nutrient conditions [24–26]. Yeast cells can catabolize many carbon sources, but prefer to use glucose if it is available. When glucose is present in the environment, the induction of the GAL pathway is strongly repressed.

Fig 1A shows a simplified diagram of the GAL regulatory network. Gal4p is the transcription factor that can initiate the transcription of GAL metabolic genes including *GAL1*, *GAL2*, *GAL3*, and *GAL80* [27]. *GAL1* encodes galactokinase which phosphorylates galactose in the first step of galactose catabolism. Gal2p is a galactose permease located in the cellular membrane, which can also take up glucose with comparable binding affinity. By transporting galactose and thus increasing intracellular galactose levels, Gal2p exerts a positive feedback on the GAL regulon [28,29]. In contrast, Gal80p is a negative feedback node that sequesters Gal4p, blocking the transcription of GAL metabolic genes. Gal3p is an intracellular galactose sensor which is activated upon binding to galactose and ATP. Activated Gal3p then forms a complex with Gal80p and relieves the sequestration of Gal4p by Gal80p [30,31]. Another important

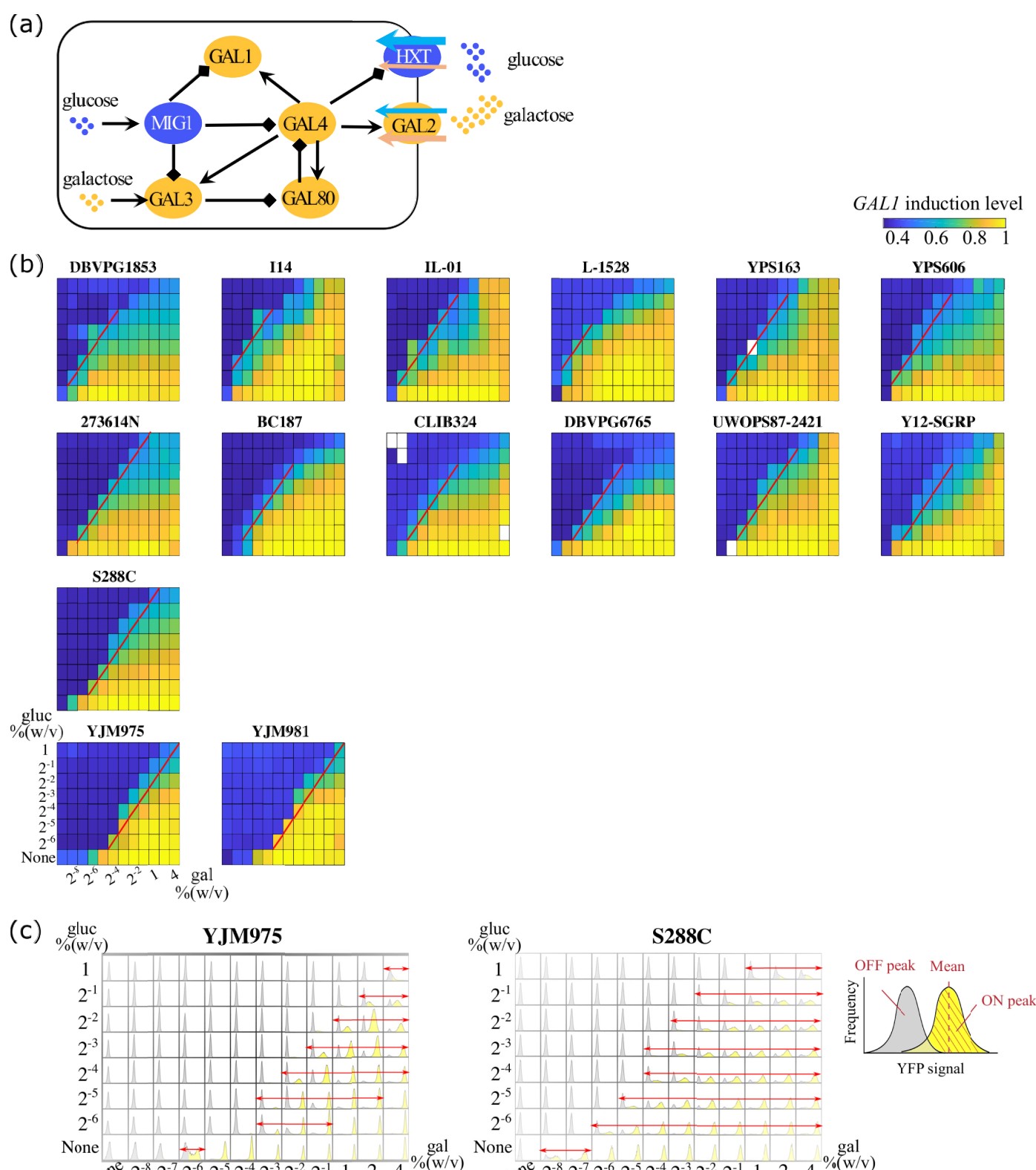

**Fig 1. Natural yeast isolates exhibit diversity in GAL pathway induction profiles.** (a) The galactose metabolic gene regulatory network in yeast. (b) The induction profiles of natural yeast isolates cultured in combinations of glucose and galactose. Each subplot represents the induction profile of the specific strain indicated as the title. Different combinations of nutrient concentrations were given to the yeast as indicated by the x and y axes. The color indicates the induction levels of

galactokinase *GAL1* in each combination of the concentrations. The induction levels were normalized to the maximum induction level of each strain. White color represents missing data. The decision front was defined as the contour line of half-maximum induction level and highlighted with red lines. From the top row to the bottom row, the decision front gradually shifts rightward, towards a higher galactose concentration. (c) Histograms showing difference in the range of unimodality and bimodality between induction profiles of YJM975 and S288C. Red arrows indicate nutrient conditions where strains exhibited bimodality. Grey shades represent OFF peaks where yeast cells did not induce the GAL pathway, whereas yellow shades represent ON peaks where yeast cells induced the GAL pathway. A magnified diagram is shown on the right. For each nutrient condition, we calculated the mean induction level of GAL-induced subpopulation (ON peak position) and the mean induction level of GAL-repressed subpopulation (OFF peak position). Note that (c) is showing the same data as YJM975 and S288C in (b). The color in (b) corresponds to the induction levels of the ON peaks where cells show bimodality or unimodal ON, and corresponds to the induction levels of the OFF peaks where cells show unimodal OFF.

regulator is *MIG1* (Multicopy Inhibitor of Galactose gene expression), which represses the transcription of GAL metabolic genes when glucose is present. The phosphorylation state of Mig1p decreases with increasing glucose levels, resulting in the transcriptional inhibition of *GAL1*, *GAL3*, and *GAL4* [32–34]. In addition to Gal2p, the hexose transporter family (HXT) also transports sugars, including galactose, into the yeast cytoplasm. The members of this family (HXT1-HXT17) have varied binding affinity for glucose, galactose and other hexoses [35–38].

In previous work, we observed significant phenotypic variation in the behavior of the GAL pathway in three different yeast isolates [39]. Here, we expand this finding to 15 isolates and explore the basis of this variation. To do this, we modeled the dynamics of GAL pathway components based on experimental observations, and determined how each parameter in our model affects GAL induction in response to glucose and galactose stimulation. We found that changes in the behavior of Gal3p, Gal80p, and HXT were sufficient to explain the observed phenotypic variation.

## Results

### Natural yeast isolates exhibit diversity in GAL pathway induction profiles

It has long been believed that when yeast cells are cultured in mixtures of glucose and galactose, the expression of the enzymes and transporters specialized for galactose utilization is repressed until glucose concentration drops below a threshold. By investigating yeast GAL pathway induction in more detail using modern high-throughput techniques, Escalante-Chong et al. found that this is not the case. Instead, yeast cells decide to induce the GAL pathway in response to the ratio of the external concentrations of galactose and glucose [39]. This initial study characterized the ratiometric response in three strains, S288C, BC187, and YJM978. Each showed the same type of response, although the nutrient conditions required for GAL pathway induction varied from one strain to another. Inspired by this observation, we set out to explore the induction profiles in other yeast strains that originated from a range of ecological niches.

We cultured 15 natural yeast isolates in a range of combinations of glucose and galactose mixed in different proportions. We introduced into each strain a reporter in which the promoter of galactokinase, *GAL1*, drove the expression of yellow fluorescent protein (YFP). By measuring the distribution of fluorescent intensity in individual cells using fluorescence-activated cell sorting (FACS), we quantified the induction levels of *GAL1* in these yeast strains. *GAL1* induction is a good surrogate for the induction of the pathway as a whole. Intriguingly, we found that each yeast isolate exhibited a different induction profile in response to identical nutrient signals (

B). To quantitatively represent the phenotypic variation among these natural yeast isolates, we defined a parameter, the "decision threshold", that describes the nutrient conditions at which yeast isolates show half maximum induction of the GAL pathway (compared to its

maximal expression in the 96 different conditions in which the isolate was queried). As the decision threshold depends on both glucose and galactose, each isolate has many "decision thresholds". These decision thresholds are linked together by the red lines in Fig 1B; we term this curve the "decision front". The decision fronts are essentially contour lines that join nutrient conditions which give rise to equal induction levels of the GAL pathway. Galactose activates the pathway via a positive feedback, whereas glucose inhibits the pathway through carbon catabolite repression. As a result, the induction level is approximately proportional to galactose concentration, while inversely proportional to glucose concentration. This leads to the decision front appearing as a straight contour line in log-log space. As different strains have various sensitivities to galactose induction and glucose inhibition, each strain has a characteristic decision front, representing the concentration ratio required to induce the GAL network in that strain. The yeast isolates we studied are arranged from the top row to the bottom row in Fig 1B such that their decision fronts shifted from left to right, i.e. from low galactose concentrations required for induction towards higher required galactose concentrations at the same level of glucose.

The decision threshold was not the only feature which varied between strains, bimodal range (the galactose concentration range within which the yeast population exhibited bimodality) differed amongst the natural yeast isolates. For example, the bimodal regions of the strains YJM975 and S288C are shown in Fig 1C. Glucose concentrations are shown from high to low, from the top row to the bottom row, while galactose levels are shown from low to high, from the left column to the right column. The histograms of *GAL1* induction levels are shown for 96 sugar combinations. Induction is highest in the lower right corner, and the bimodal region is highlighted with red arrows. YJM975 showed a relative narrow bimodal range: even at very low glucose concentration (0.016% gluc), the GAL pathway was completely repressed until galactose level increased to a medium-high concentration (0.125% gal). At 1% gal and 0.016% gluc all YJM975 cells had induced *GAL1*. In contrast, S288C showed much broader regions of bimodality.

To explore how bimodality might vary in different strains, we built a deterministic ODE model (see Material and Methods). We used conventional Michaelis-Menten kinetics and Hill equations to simulate the dynamics of canonical components of the GAL pathway, including GAL1p, Gal2p, Gal3p, Gal4p, Gal80p, and the complexes they form. We also included Mig1p and HXT, which regulate the GAL pathway from outside the network, and the levels of intracellular glucose and galactose. A major difference between our model and existing GAL models [23,24,26] is that we included the competitive binding of glucose and galactose to shared transporters. As reported in previous work, Gal2p has comparable binding affinity to glucose and galactose[38], and both HXT11 and HXT17 can transport galactose [37,40]. Thus, the levels of galactose and glucose within the cell depend on the external concentrations of both sugars, and the behavior of the different sugar transporters.

We used our ODE model to simulate the dynamics of GAL pathway components, then compared the simulated results with our experimentally measured data. We chose this method because the bimodality of the GAL pathway has been shown to arise from an underlying bistability [22,23] and a deterministic ODE model is much less computationally expensive than stochastic simulations. To identify regions where bistability (and hence bimodality) exists, we started from either high galactose or no galactose initial conditions and allowed the system to evolve towards steady state. These simulations recapitulated the three basic responses we saw experimentally: 1) an area where all cells repressed the GAL pathway (unimodal OFF), 2) an area where all cells induced the GAL pathway (unimodal ON), and 3) a mixed population with some cells repressing and some cells inducing the pathway (bimodal) (S1 Fig).

## Model validation of synthetic deletion

To help constrain the number of parameters in our GAL model, we fit our model both to the wild-type response of an S288C strain and to two GAL pathway mutants in this strain (S288C$^{gal80\Delta}$ and S288C$^{mig1\Delta}$). We reasoned that simultaneously fitting the model to all three strains would reduce the risk of over-fitting and might identify parameters that were better at fitting natural variation. We genetically knocked out *GAL80* and *MIG1* separately in the S288C strain, then characterized the responses of the S288C$^{gal80\Delta}$ and S288C$^{mig1\Delta}$ strains to mixtures of glucose and galactose, as for the wild-type strain. We used our mathematical model to simulate the induction levels of the mutant strains, varied the parameters to minimize mean squared deviation, and searched for parameter values that would fit the experimental data of all three strains. The objective function was defined as

$$Obj = \sum_{i=1}^{n}(y_i^{high} - f(x_i^{high}, \beta))^2 + \sum_{i=1}^{n}(y_i^{low} - f(x_i^{low}, \beta))^2 \tag{1}$$

where *i* stands for the index of 96 nutrient conditions, $x_i^{high}$ and $x_i^{low}$ represent the initial conditions to reach high and low steady states, respectively, $\beta$ stands for a set of parameter values, and $y_i^{high}$ and $y_i^{low}$ represent the experimentally determined mean induction levels of ON-peaks (GAL-induced subpopulation) and OFF-peaks (GAL-repressed subpopulation, Fig 1C), respectively.

We plotted parameter values that can fit wildtype and mutant S288C data separately, and compared the parameter value range between these three strains (Fig 2A). As we expected, most parameters showed overlaps between wildtype and mutant strains, and the most significant differences came from the basal synthesis rate of Gal80p (a80), the maximum synthesis rate of Gal80p (ag80), the synthesis rate of Mig1p (aR), and the *GAL1* transcriptional threshold for Mig1p inhibition (KR1). This is expected, since these parameters respond directly to the genetic changes that we made in S288C$^{gal80\Delta}$ and S288C$^{mig1\Delta}$. Thus, the parameters describing the behavior of the wildtype strain were consistent with those for the genetic mutants, so that our mechanistic model was able to capture the induction profiles of wildtype and mutants simultaneously. Fig 2B shows the results of one 'best-fit' parameter set. We used identical parameter values in all the three strains, except for parameters corresponding to the synthesis rates of genetic mutants, i.e. a80, ag80, and aR. Although there are quantitative differences in the GAL induction levels for all three strains between the simulations and experiments ($Obj^{WT}$ = 7.59, $Obj^{mig1\Delta}$ = 7.80, $Obj^{gal80\Delta}$ = 14.88), and the overall fitting was not as good as only fitting to wildtype S288C data (as shown in S1A Fig, *Obj* = 6.76), only fitting to S288C$^{mig1\Delta}$ data (as shown in S1B Fig, *Obj* = 2.56), or only fitting to S288C$^{gal80\Delta}$ data (as shown in S1C Fig, *Obj* = 2.06), this parameter set captured the ratiometric response in the wildtype S288C, the elevated induction levels in S288C$^{mig1\Delta}$, and the glucose threshold sensing in S288C$^{gal80\Delta}$ (*GAL1* induction levels only respond to glucose titration and are insensitive to changes in galactose concentration, as data of S288C$^{gal80\Delta}$ showing in Fig 2B). Meanwhile, simultaneously fitting to three strains data avoids overfitting to a specific strain and retains the generality to recapitulate common characteristics shared by natural yeast isolates. Hence, we stuck with this parameter set as starting parameters in our following investigations.

## *In silico* survey of parameters that can affect bimodal range

We next tested whether perturbing parameter values in our model could recapitulate the observed variation in the bimodal range across natural yeast isolates. We defined three metrics, $\delta_{ON}$, $\delta_{full}$, and $\delta_{level}$ (Fig 3A), to quantify the difference in bimodal range and expression level

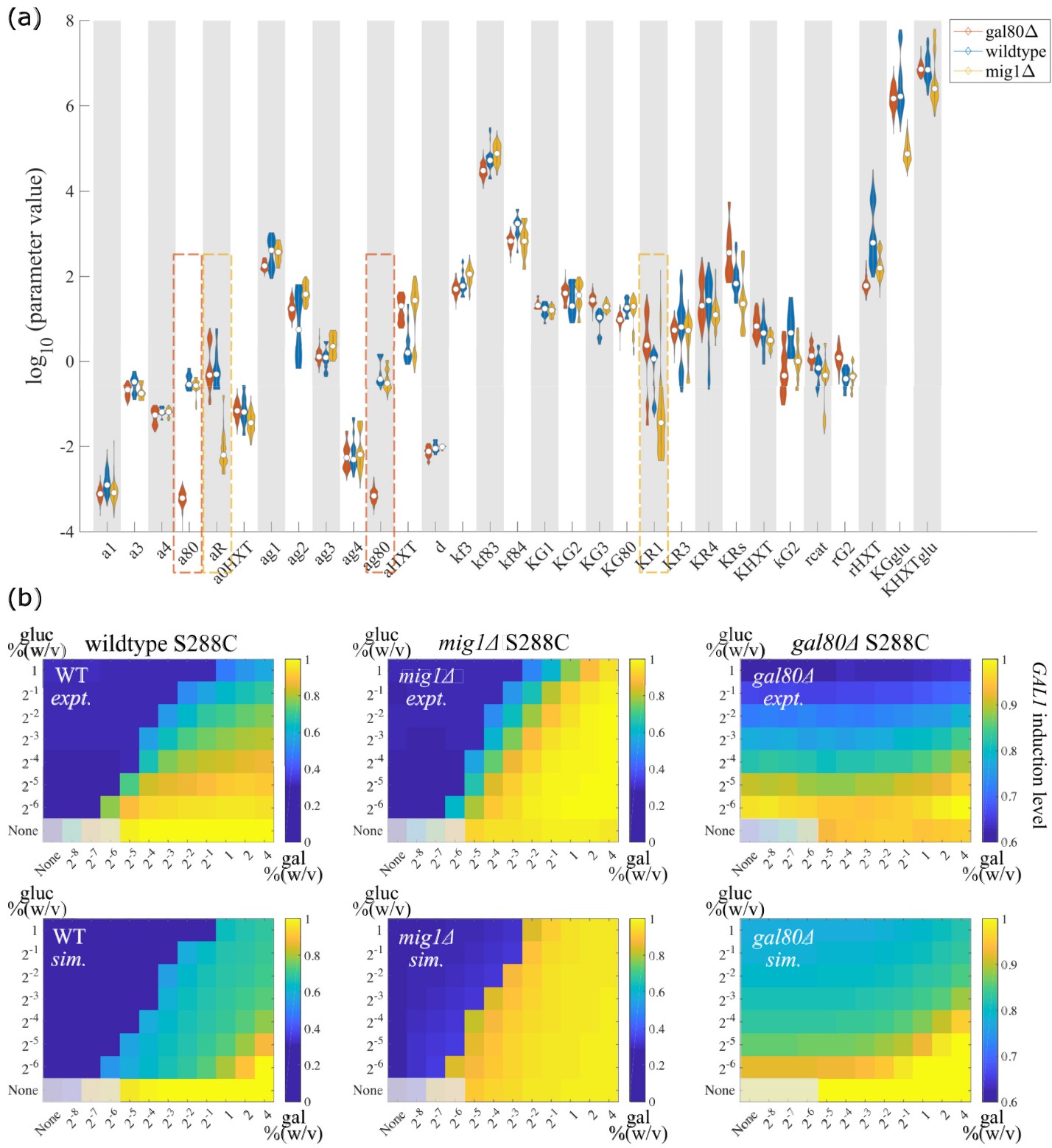

**Fig 2. Using mechanistic model to capture phenotypes in wildtype and mutant S288C strains.** (a) Parameter choices are consistent in wildtype and synthetic deletion strains of S288C. The horizontal axis shows the free parameters that we allowed to vary in the fitting for wildtype S288C (wildtype), for *gal80Δ* synthetic deletion strain (*gal80Δ*), and for *mig1Δ* synthetic deletion strain (*mig1Δ*). The vertical axis shows the parameter values that were able to fit wildtype and mutant S288C strains separately. Most parameter values overlapped among the three strains, except parameters corresponding to synthetic deletions, including a80, the basal synthesis rate of Gal80p, ag80, the maximum synthesis rate of Gal80p (both highlighted in red dashed boxes), and aR, the synthesis rate of Mig1p, KR1, *GAL1* transcriptional

threshold for Mig1p inhibition (both highlighted in yellow dashed boxes). (b) Using the best-fit parameters to simultaneously capture the phenotypes of wildtype and mutant S288C strains. Top row shows the experimentally determined induction profiles of wildtype S288C, *mig1Δ* deletion S288C, and *gal80Δ* deletion S288C, from left to right respectively. Bottom row shows the simulation results for wildtype S288C, *mig1Δ* deletion S288C, and *gal80Δ* deletion S288C. Different combinations of nutrient concentrations were given to the yeast as indicated by the x and y axes. Color codes for the induction levels of galactokinase Gal1p. The induction levels were normalized to the maximum induction level of each strain that was measured experimentally. The parameter values used to simulate the three strains were identical, except that the synthesis rate of Mig1p was set to zero in *mig1Δ* deletion S288C, and the synthesis rate of Gal80p was set to zero in *gal80Δ* deletion S288C. We grayed out the four leftmost squares in the bottom row of each heatmap as those have very poor to no growth.

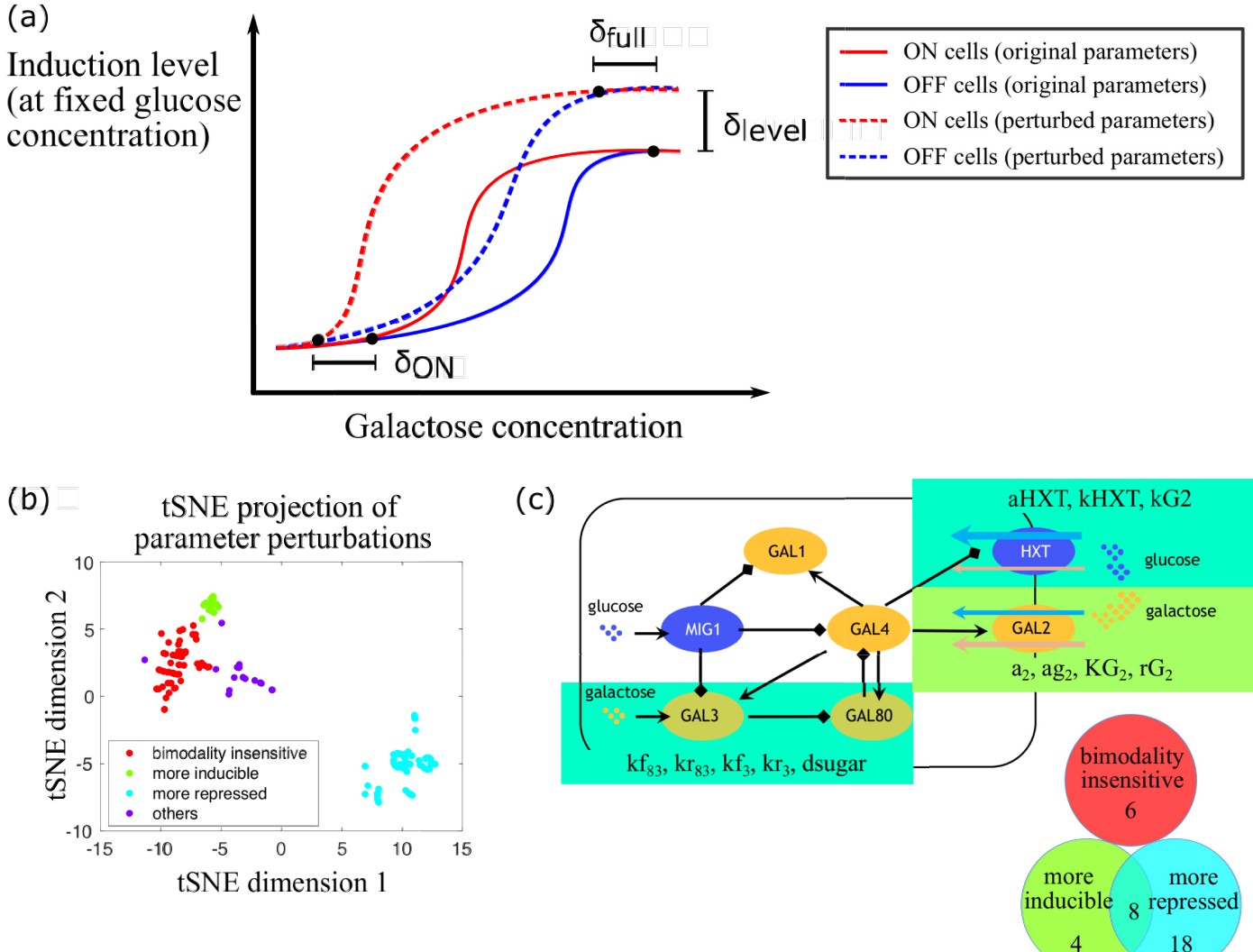

**Fig 3. *In silico* survey of parameters that can affect bimodal range.** (a) Illustration of the induction metrics we inspected. Bimodal range is affected by inherent difference between yeast strains which we modeled by perturbing parameter values, and such yeast cells exhibited bimodality at lower galactose concentrations. The difference in the galactose concentrations where bimodality emerged is represented by $\delta_{ON}$. Similarly, the difference in the galactose concentrations where bimodality vanished is represented by $\delta_{full}$. $\delta_{level}$ represents the difference in the steady state induction levels. (b) t-distributed stochastic neighbor embedding (t-SNE) visualization of parameter perturbation clusters identified according to their effect on bimodal range variation. Minkowski distance metric was used, loss = 0.080. (c) When increasing and decreasing parameter values, 8 parameters made the pathway both more inducible and more repressed, and 4 parameters related to Gal2p made the pathway more inducible, these parameters with their associated biological implications are highlighted in color boxes. Moreover, 18 parameters made the pathway more repressed, and 6 parameters showed minor effect on bimodal range.

resulting from changes in one or more parameters. We then simulated yeast cell behavior with different parameter sets, in each case starting with either high galactose (ON cells, red lines in Fig 3A) or no galactose (OFF cells, blue lines in Fig 3A) initial conditions and allowing the system to evolve towards steady state. The region in which the two simulations diverge is the region of bimodality. When comparing two sets of parameter choices, $\delta_{ON}$ represents the difference in the galactose concentration where bimodality emerged, at a fixed glucose concentration. A positive value of $\delta_{ON}$ means that the new parameter set caused bimodality to emerge at higher galactose concentrations, i.e. showed stronger repression of the GAL pathway, whereas a negative value of $\delta_{ON}$ means that the GAL pathway was easier to induce. Similarly, $\delta_{full}$ represents the difference in the galactose concentration at which bimodality vanished because all cells are fully induced, again given a fixed glucose concentration. $\delta_{level}$ quantifies the difference in induction levels at full induction.

To study how each parameter can affect the bimodal range, we next perturbed free parameters in our model one at a time, either increasing or decreasing the best-fit parameter values we obtained from fitting S288C wt, S288C$^{gal80\Delta}$ and S288C$^{mig1\Delta}$ by 2-fold, 10-fold, or 100-fold. For each parameter condition, we fixed the glucose concentration and simulated titrating galactose from low to high or high to low concentrations to determine the bimodal region. This was repeated for eight glucose concentrations (including zero glucose). We did not include the Hill coefficients in the model as variable parameters, since we wanted to maintain the basic structure of the control system.

These simulations resulted in a matrix recording how each parameter perturbation affected $\delta_{ON}$, $\delta_{full}$, and $\delta_{level}$ at each of the eight glucose concentrations. After performing these simulations, we clustered parameter perturbations based on their influence on bimodal range and induction levels (S2 Fig). We found that the parameter perturbations can be roughly divided into three groups, 'more inducible', 'more repressed', and 'bimodality insensitive'. The 'more inducible' group of perturbations (S3 Fig) enabled bimodality to emerge at lower galactose concentrations (the galactose concentrations where bimodality emerged were decreased by ≥8-fold in multiple fixed glucose concentrations) and also increased the induction levels of the GAL pathway. In contrast, 'more repressed' perturbations (S4 Fig) caused bimodality to emerge at higher galactose concentrations (bimodality emerged at ≥8-fold higher galactose concentrations), or even eliminated bimodality completely (the pathway degenerated to a unimodal OFF system), and decreased the induction levels of the GAL pathway. There were also a number of parameter perturbations that had only a small effect on $\delta_{ON}$, $\delta_{full}$, and $\delta_{level}$ (typically $\delta_{ON}$, $\delta_{full}$ of ~0 to ~2-fold, and $\delta_{level}$ of ~0). These are designated the 'bimodality insensitive' group (S5 Fig). A few parameter perturbations did not fall into any of these categories ('others' group). The effect of parameter perturbations on $\delta_{ON}$, $\delta_{full}$, and $\delta_{level}$ individually is shown in S6 Fig.

The resulting perturbation matrix was used to visualize the effects of parameter perturbation on the bimodal range in two dimensions using t-distributed stochastic neighbor embedding (Fig 3B). The cluster results suggested that among the 36 free parameters in our model, there were 8 that were important in bimodal range variation and that could either cause the pathway to become more inducible, or more repressed (Fig 3C). These included the binding (or unbinding) rate between Gal80p and Gal3p (named kf83 or kr83), the binding (or unbinding) rate between Gal3p and galactose (kf3 or kr3), the turnover rate of intracellular sugars (dsugar), the synthesis rate of hexose transporters (aHXT), and the uptake rates of HXT and Gal2p (kHXT and kG2, respectively). We also found that changes in parameters related to Gal2p, including the basal and maximum synthesis rates of Gal2p (a2 and ag2), the transcriptional activation threshold by Gal4p (KG2), and the relative binding affinity between glucose and galactose to Gal2p (rG2) made the pathway more inducible when their parameter values

were increased (for a2 and ag2) or decreased (for KG2 and rG2). Changes in the other direction did not clearly make the pathway more repressed, however. There were 6 parameters that did not affect bimodal range (bimodality insensitive) and 18 parameters that would cause the pathway to be more repressed when their parameter values were perturbed. Representative examples of each cluster in Fig 3B are shown in S7 Fig. All the parameters that showed significant effects similar to the natural variation of the bimodal range were related to Gal3p, Gal80p, and HXT.

## Decision front shift and phenotype switching between natural yeast isolates

Having identified parameters that affected the bimodal range, we wished to determine whether these parameters alone are sufficient to switch the *in silico* phenotype from one strain to another. In particular, we wanted to know whether the decision front could be shifted from one strain's phenotype to another's by tuning the driving factors of bimodal range variation.

We used the best-fit parameter set developed to simulate wildtype and mutant S288C phenotypes as our starting parameter values, and set out to determine how much improvement in the fit to the phenotype of each natural yeast isolate (see Fig 1B) can be accomplished by tuning one parameter at a time. We calculated the following metric for every query strain and every parameter:

$$\Delta Obj = \frac{Obj(strain_i, param) - min(Obj(strain_i, \tilde{param}_j))}{Obj(strain_i, param)} \tag{2}$$

$Obj(strain_i, param)$ is the objective function calculated for the $i_{th}$ yeast strain, using the starting parameter values, $Obj(strain_i, \tilde{param}_j)$ is the objective function calculated by tuning the $j_{th}$ parameter value and keeping all the other parameter values unchanged. Among all the tuned parameter values, the one that leads to the minimum objective function was then compared to the starting objective value to calculate the improvement, $\Delta Obj$, i.e., $\Delta Obj$ quantifies the extent to which the fitting of a query strain was improved, which is shown in Fig 4 as a color-coded matrix. Each row represents one of our 15 experimentally measured yeast strains, and each column represents one of 45 parameters in our model. The deeper the color, the more the fitting was improved. To give an intuitive sense of how changing one parameter improves fitting of the actual data better than changing the others, we show the fitting of I14 strain by singly tuning rHXT (S8A Fig), by singly tuning KRs (S8B Fig), and by singly tuning KG1 (S8C Fig), with fitting improvement = 90%, fitting improvement = 60%, and fitting improvement = 60%, respectively. More examples showing how singly tuning one parameter could improve the fitting can be found in S8D, S9, and S10 Figs.

We subsequently used heatmap to visualize the induction profiles of the 7 strains that showed a fitting improvement ≥70% in Fig 4. The comparison between the simulation results and the experimental data is shown in Fig 5. Each subplot shows experimental data for a single strain at the top (*expt.*) and results from the simulation that attempts to mimic this strain at the bottom (*sim.*). The concentration ratio of glucose to galactose that was required for the specific strain to induce the GAL pathway (induction ratio) is indicated at the upper left, and the parameter modification to achieve phenotype switching compared to the starting parameter values is indicated at the lower left.

We found that the phenotypes of three strains, I14 (Fig 5A), YPS163 (Fig 5B), and UWOPS87-2421 (Fig 5C), could be accessed from the reference strain S288C by singly tuning rHXT. This parameter measures the ratio of glucose and galactose binding affinity to HXT. We found that decreasing rHXT (corresponding to increasing relative affinity of galactose and glucose to HXT) in the simulations shifted the decision front leftward, towards a higher

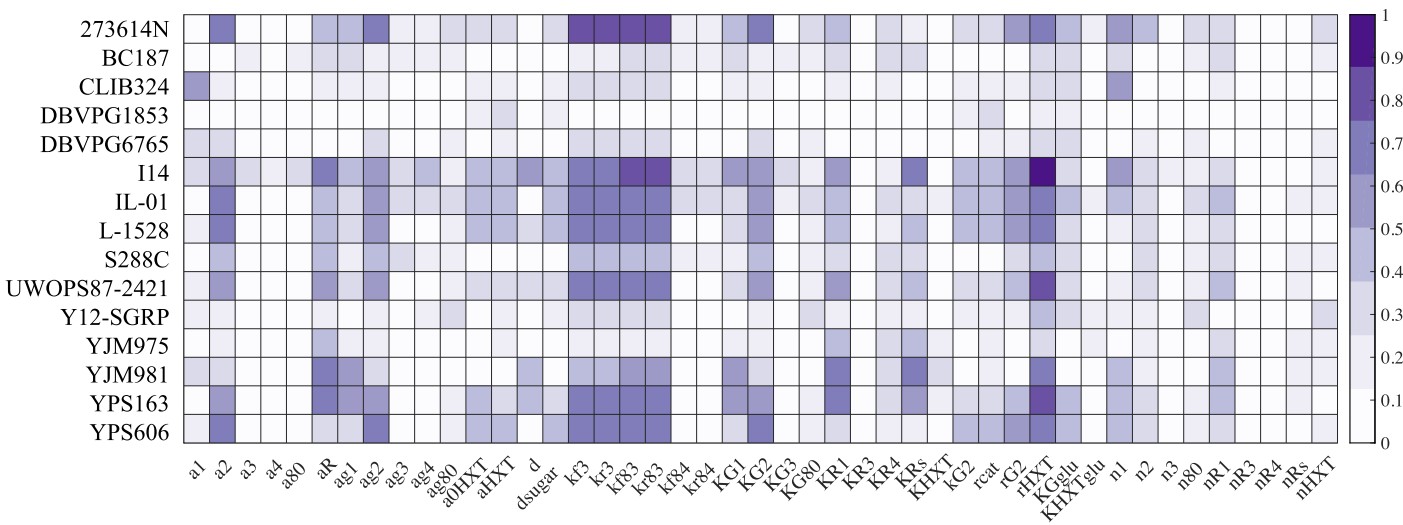

**Fig 4. Fitting improvement matrix.** A color-coded matrix showing to what degree each parameter modification could improve the fitting of each natural yeast isolate. Each row represents one of fifteen experimentally measured yeast strains, each column represents one of forty-five parameters in our model. The deeper the color means the more the fitting was improved.

glucose/galactose ratio. This is consistent with our earlier finding that competition between sugars for transporter binding could give rise to a ratiometric signal response like the one observed in the GAL pathway, and that the relative binding affinity of the competing carbon sources for the communal transporter sets the concentration ratio for the induction of the pathway [41]. The induction ratio of I14 was 8 (glucose over galactose), meaning that the induction level of the GAL pathway was higher than half maximum at nutrient conditions of 0.031% glucose with 0.004% galactose ($\frac{gluc}{gal} = 8$), and 0.063% glucose with 0.008% galactose ($\frac{gluc}{gal} = 8$). In contrast, the induction ratios of YPS163 and UWOPS87-2421 were 4 and 2, respectively, indicating they require lower glucose concentrations (or higher galactose concentrations) in the environment to induce the GAL pathway than does I14.

For three additional strains, L-1528 (Fig 5D), IL-01 (Fig 5E), and 273614N (Fig 5F), the key parameter was kf83, the binding rate between Gal3p and Gal80p. These three strains have induction ratios of 8, 4, and 2, respectively, while the induction ratio of S288C is 1. We found that increasing kf83 in the simulations was sufficient to alter the S288C induction ratio, shifting it towards higher glucose/galactose ratios to different extents, and matching the phenotypes of the test strains. As Gal80p serves as the negative feedback node in the GAL network, increasing the binding rate between Gal3p and Gal80p is equivalent to lowering the strength of GAL80-mediated negative feedback which further results in the increased activity of the GAL network. Consistent with this, another study that replaced the promoters of *GAL80* in *Saccharomyces cerevisiae* with their counterparts from *Saccharomyces paradoxus* also observed higher network inducibility levels, reinforcing the idea that *GAL80* influences inducibility[42].

The final strain in our set of 7, YPS606 (Fig 5G) provides insight into a different network parameter. The phenotype of YPS606 can be simulated by increasing kf3, the parameter reflecting the binding rate between galactose and Gal3p. This shifts the decision front leftwards, reaching an induction ratio of 4 (Fig 5G). In order to validate whether these key parameters identified by our model indeed affect GAL induction in yeast cells, we performed a *GAL3* allele swap experiment, i.e., introducing the *GAL3* allele from YPS606 to a S288C^{gal3Δ} genetic background, and measured the induction profiles in the same experimental setting (Fig 5H).

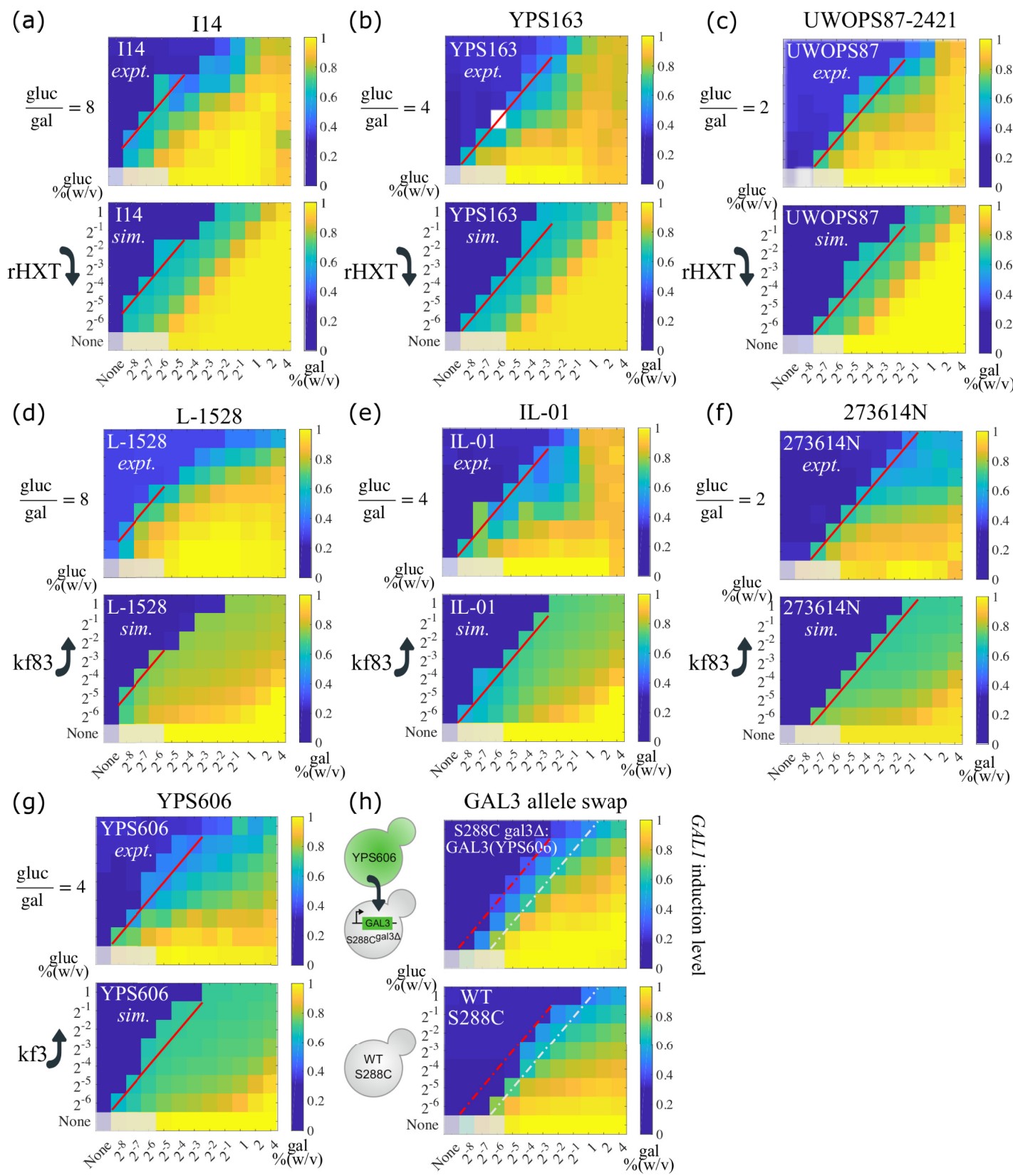

**Fig 5. Decision front shift and phenotype switching between natural yeast isolates.** Seven query strains achieved phenotype switching from the reference strain S288C (fitting improvement was greater than 70%) when singly tuning one parameter of mechanistic sources underlying natural variation. The strains are indicated at the top of each subplot. (a)—(g) Comparison between experimental data (on the top) and simulated results (at the bottom) of the same strain is shown, with color coding induction levels of the GAL pathway and decision front highlighted in red lines. White color represents missing data. The concentration ratio of glucose to galactose that is required for the specific strain to induce the GAL pathway is indicated upper left, and the parameter modification to achieve phenotype switching is indicated lower left. We grayed out the four leftmost squares in the bottom row of each heatmap as those have very poor to no growth. (h) Induction levels of *GAL3* allele swap experiment. *GAL3* allele from YPS606 was introduced into a S288C$^{gal3\Delta}$ genetic background (upper panel). Compared to wildtype S288C (lower panel), this engineered strain has a decision threshold shifted toward a higher glucose/galactose ratio. The red dash line indicates the decision front predicted by the model. The gray dash line indicates the decision front in wildtype S288C.

Indeed, the resulting engineered strain has a decision threshold shifted toward a higher glucose/galactose ratio.

## Discussion

Numerous studies have been carried out to uncover the genetic and molecular basis of natural variation using *S. cerevisiae* as a model system. The rich ecological history that exists within yeast species has shaped different yeast strains with various behaviors under interacting genetic and environmental driving forces. Here we studied the natural variation in the induction profiles of the GAL pathway when yeast cells are stimulated by coexisting but conflicting galactose and glucose nutrient signals, the former one activating the GAL pathway whereas the latter one inhibiting the pathway. Identifying the mechanisms underlying such natural variation will deepen our understanding of how microbes cope with complex and fluctuating environments and how cells make decisions in response to input signals. Previous studies have shown that natural variation in the lag-phase in yeast diauxic growth reveals a cost-benefit tradeoff [25], and the bimodality in response to nutrient signals within a yeast population promotes anticipation of environmental shifts [26]. The natural variation in the steady state induction levels we observed in this research could be another manifestation of the variation in yeast nutrient strategies.

Most studies regarding quantitative traits used experimental techniques such as perturbation and deletion screens to identify gene functions. We approached this problem from a complementary computational perspective. Based on existing GAL models and incorporating the core modules we found were responsible for ratiometric response that was observed in the GAL pathway, we obtained a mechanistic model which we first validated through the simultaneous fitting of wildtype and mutant S288C strain data.

A key question here was whether our models were sufficient to be able to recapitulate natural variation and predict the underlying mechanistic changes that underlie this variation. Indeed, for each trait we explored, we could recapitulate the phenotypic behavior with the modulation of a single gene. But often more than one single gene could elicit the same behavior. While not as straightforward as a system in which one gene control one part of each phenotype, the result conceptually makes sense. For example, any gene that can affect the amount of the complex formed by Gal80p and Gal 3p can shift the induction ratio. This can be achieved by affecting the level of Gal3p, Gal80p or their interaction. This shared behavior was easily observed when we clustered the parameter perturbations based on their effect on phenotypes such as the bimodal range and induction level; parameters with biological implications related to the galactose sensor Gal3p, the negative feedback node Gal80p, and the hexose transporter HXT had the most explicit influence on bimodal range.

Since the experimental approaches to determine genetic sources of natural variation remains laborious, our results suggest that mathematical modeling and computational studies can be useful to help pinpoint mutant studies for further experimental analysis. Hopefully,

with the growing availability of data in other systems, similar models and analysis could explain phenotypic variations that are wide-spread in biological systems.

## Materials and methods

The source code for mathematical modeling and *in silico* simulations is available at https://github.com/JiayinHong/Natural-variation-in-yeast-GAL-pathway-induction.

### Strains and media

Strains were obtained as described in [25,43]. Strains used in this study can be found in S1 Table. All strains were homozygous diploids and prototrophic. Strains were assayed in a gradient of glucose (1% to 0.016% by twofold dilution) in combination with a gradient of galactose (4% to 0.004% by twofold dilution). All experiments were performed in synthetic minimal medium, which contains 1.7g/L Yeast Nitrogen Base (YNB) (BD Difco) and 5g/L ammonium sulfate (EMD), plus D-glucose (EMD) and D-galactose (Sigma). Cultures were grown in a humidified incubator (Infors Multitron) at 30˚C with rotary shaking at 230rpm (tubes and flasks) or 999rpm (600uL cultures in 1mL 96-well plates).

### Flow cytometry assay

GAL induction experiments were performed in a twofold dilution series of glucose concentration, from 1% to 0.016% w/v, with a twofold dilution series of galactose concentration, from 4% to 0.004% w/v. To start an experiment, cells were struck onto YPD agar from -80˚C glycerol stocks, grown to colonies, and then inoculated from colony into YPD liquid and cultured for 16–24 hours. The optical density (OD600) of the cultures was measured on a plate reader (PerkinElmer Envision), and once OD600 reached 0.1, cells were then washed once in solution consisting of 0.17% Yeast Nitrogen Base and 0.5% Ammonium Sulfate. Washed cells were diluted 1:200 into glucose + galactose gradients in 96-well plates (500uL cultures in each well) and incubated for 8 hours. Then, cells were processed by washing twice in Tris-EDTA pH 8.0 (TE) and resuspended in TE + 0.1% sodium azide before transferring to a shallow microtiter plate (CELLTREAT) for measurement.

Flow cytometry was performed using a Stratedigm S1000EX with A700 automated plate handling system. Data analysis was performed using custom MATLAB scripts, including Flow-Cytometry-Toolkit (https://github.com/springerlab/Flow-Cytometry-Toolkit). *GAL1*pr-YFP expression was collected and the induced subpopulations (ON peaks) for each concentration of sugars was determined as shown previously in Escalante et al. [39]. We collected the mean induction levels of the GAL-induced subpopulations (ON peak positions) and the mean expression levels of the GAL-repressed subpopulations (OFF peak positions) as experimental data for subsequent comparison with simulated results.

### Mathematical modeling

We constructed an ODE (Ordinary Differential Equations) model of the GAL gene regulatory network based on the interactions shown in Fig 1A. This model was able to provide explanations for experimental data and insights about the mechanistic sources underlying natural variation in yeast GAL pathway induction profiles. We kept some simplifications that had been made to the GAL pathway in existing models [23,26]: dimerization of Gal4p and Gal80p was not modeled, subcellular localization of the GAL proteins was not considered, and a glucose repressor, $R_s$ (for example Mig1p) transcriptionally represses *GAL1*, *GAL3*, and *GAL4*. Compared to existing GAL models, the major modifications to our model including the following:

(*i*) we introduced sugar uptake through membrane transporters including galactose permease, Gal2p and hexose transporters, HXT, (*ii*) we used one species HXT to model the integrative transportation capacity through HXT1-HXT17, (*iii*) we modeled intracellular galactose and glucose, incorporating sugar uptake and turnover. The model equations based on these assumptions are

$$\frac{dG_1}{dt} = a_1 + ag_1 \cdot \frac{G_4^{n_1}}{G_4^{n_1} + K_{G1}^{n_1}} \cdot \frac{K_{R1}^{n_{R1}}}{K_{R1}^{n_{R1}} + R_s^{n_{R1}}} - d \cdot G_1 \tag{3}$$

$$\frac{dG_2}{dt} = a_2 + ag_2 \cdot \frac{G_4^{n_2}}{G_4^{n_2} + K_{G2}^{n_2}} - d \cdot G_2 \tag{4}$$

$$\frac{dG_3}{dt} = a_3 + ag_3 \cdot \frac{G_4^{n_3}}{G_4^{n_3} + K_{G3}^{n_3}} \cdot \frac{K_{R3}^{n_{R3}}}{K_{R3}^{n_{R3}} + R_s^{n_{R3}}} - kf_3 \cdot gal \cdot G_3 + kr_3 \cdot G_3^* - d \cdot G_3 \tag{5}$$

$$\frac{dG_3^*}{dt} = kf_3 \cdot gal \cdot G_3 - kr_3 \cdot G_3^* - kf_{83} \cdot G_3^* \cdot G_{80} + kr_{83} \cdot C_{83} - d \cdot G_3^* \tag{6}$$

$$\frac{dG_{80}}{dt} = a_{80} + ag_{80} \cdot \frac{G_4^{n_{80}}}{G_4^{n_{80}} + K_{G80}^{n_{80}}} - kf_{83} \cdot G_3^* \cdot G_{80} + kr_{83} \cdot C_{83}$$

$$-kf_{84} \cdot G_4 \cdot G_{80} + kr_{84} \cdot C_{84} - d \cdot G_{80} \tag{7}$$

$$\frac{dG_4}{dt} = a_4 + ag_4 \cdot \frac{K_{R4}^{n_{R4}}}{K_{R4}^{n_{R4}} + R_s^{n_{R4}}} - kf_{84} \cdot G_4 \cdot G_{80} + kr_{84} \cdot C_{84} - d \cdot G_4 \tag{8}$$

$$\frac{dC_{83}}{dt} = kf_{83} \cdot G_3^* \cdot G_{80} - kr_{83} \cdot C_{83} - d \cdot C_{83} \tag{9}$$

$$\frac{dC_{84}}{dt} = kf_{84} \cdot G_4 \cdot G_{80} - kr_{84} \cdot C_{84} - d \cdot C_{84} \tag{10}$$

$$\frac{dHXT}{dt} = a_{0HXT} + a_{HXT} \cdot \frac{K_{HXT}^{n_{HXT}}}{K_{HXT}^{n_{HXT}} + G_4^{n_{HXT}}} - d \cdot HXT \tag{11}$$

$$\frac{dR_{tot}}{dt} = a_R - d \cdot R_{tot} \tag{12}$$

$$R_s = \frac{gluc^{n_{Rs}}}{gluc^{n_{Rs}} + K_{Rs}^{n_{Rs}}} \cdot R_{tot} \tag{13}$$

$$\frac{dgluc_{in}}{dt} = k_{G2} \cdot G_2 \cdot \frac{gluc_{ex}}{\frac{1}{r_{G2}} \cdot gal_{ex} + gluc_{ex} + K_{Ggluc}}$$

$$+(r_{cat} \cdot k_{G2}) \cdot HXT \cdot \frac{gluc_{ex}}{\frac{1}{r_{HXT}} \cdot gal_{ex} + gluc_{ex} + K_{HXTgluc}} - d_{sugar} \cdot gluc_{in} \tag{14}$$

$$\frac{dgal_{in}}{dt} = k_{G2} \cdot G_2 \cdot \frac{gal_{ex}}{r_{G2} \cdot gluc_{ex} + gal_{ex} + (r_{G2} \cdot K_{Ggluc})} - kf_3 \cdot gal \cdot G_3 + kr_3 \cdot G_3^*$$

$$+ (r_{cat} \cdot k_{G2}) \cdot HXT \cdot \frac{gal_{ex}}{r_{HXT} \cdot gluc_{ex} + gal_{ex} + (r_{HXT} \cdot K_{HXTgluc})} - d_{sugar} \cdot gal_{in} \tag{15}$$

Eqs (3)–(5), and (7)–(8) are the dynamics of Gal1p, Gal2p, Gal3p, Gal80p, and Gal4p, respectively. Eq (6) is the dynamics of activated Gal3p that is bound by galactose and ATP. Eqs (9)–(10) are the dynamics of complexes formed by Gal80p and Gal3p, by Gal80p and Gal4p, respectively. Eq (11) is the dynamics of the integrated species HXT. Eq (12) is the dynamics of the total amount of repressors. Eq (13) is the activated repressor that is bound by glucose. Eqs (14)–(15) are the dynamics of intracellular glucose and galactose, respectively.

## Estimation and optimization of model parameters

Initial parameters for the model were estimated from experimental measurements and previous studies [24,26,35,44–48] (see parameter descriptions and values in S2 Table). We then optimized parameter values based on Metropolis-Hastings algorithm so as to reproduce the correct responses in the GAL pathway induction studies. The algorithm imitates random walks in parameter space, using a proposal density for new steps and a method for rejecting some of the proposed moves. In our optimization problems, posterior probability is the probability of the model parameters (*param*) given the experimentally measured data (*data*), in contrast with the likelihood function, which is the probability of the data given the model parameters. The two are related as follows

$$P(param|data) = P(data|param) \times \frac{P(param)}{P(data)} \propto P(data|param) \times P(param) \tag{16}$$

The likelihood function was given in the form of Gaussian noise, thus we obtained

$$P(data|param) = \frac{1}{\sigma\sqrt{2\pi}} e^{-\frac{(x-\mu)^2}{2\sigma^2}} \propto e^{-\frac{(x-\mu)^2}{\sigma^2}} \tag{17}$$

And *P*(*param*) is the prior probability distribution that we chose to closely match published estimates whenever possible. Hence, we derived the relation between the posterior probability and the prior probability as follows

$$log\, P(param|data) \propto -\frac{(x-\mu)^2}{\sigma^2} + log\, P(param) \tag{18}$$

We then used custom MATLAB codes to implement the algorithm to maximize a posterior probability, and equivalently, to minimize the objective function described in Eq (1).

To generate Fig 4, the parameters were scanned across 6 orders of magnitude centered on the starting parameter (3 orders of magnitude up to 1000-fold, 3 orders of magnitude down to 1/1000), one parameter at a time. For each parameter, 48 points were evenly sampled in logarithmic space across these 6 orders of magnitude flanking the default parameter value. The minimization was taken over the resultant objective function.

## Calculation of induction metrics

We defined three metrics, $\delta_{ON}$, $\delta_{full}$, and $\delta_{level}$, to quantify the difference in bimodal range and expression level resulting from changes in one or more parameters. We then simulated yeast cell behavior with different parameter sets, in each case starting with either high galactose (ON cells, red lines in Fig 3A) or no galactose (OFF cells, blue lines in Fig 3A) initial conditions and allowing the system to evolve towards steady state. The region in which the two simulations diverge is the region of bimodality. Specifically, we considered nutrient conditions at which the high state level was $\geq$5-fold higher than the low state level to be bimodal. When comparing two sets of parameter choices, $\delta_{ON}$ represents the difference in the galactose concentration where bimodality emerged, at a fixed glucose concentration. A positive value of $\delta_{ON}$ means that the new parameter set caused bimodality to emerge at higher galactose concentrations, for instance, from 0.0625% to 0.25% (in this case, $\delta_{ON} = log_2 \frac{0.25\%}{0.0625\%} = +2$), showing stronger repression of the GAL pathway. Conversely, a negative value of $\delta_{ON}$ means that the parameter perturbation enabled bimodality to emerge at lower galactose concentrations, say, from 0.125% to 0.0625% (in this case, $\delta_{ON} = log_2 \frac{0.0625\%}{0.125\%} = -1$), showing the GAL pathway easier to induce. Similarly, $\delta_{full}$ represents the difference in the galactose concentration at which bimodality disappeared because all cells are fully induced, again given a fixed glucose concentration. Note that for glucose concentrations at which the GAL pathway kept bimodal even at the highest measured galactose concentration, a positive value of $\delta_{full}$ denotes that the perturbation degenerated the system to unimodal OFF. $\delta_{level}$ quantifies the difference in induction levels at full induction.

$$\delta_{level} = \frac{inductionlevel'_{(perturbedparameter)} - inductionlevel_{(defaultparameter)}}{inductionlevel_{(defaultparameter)}} \quad (19)$$

A positive value of $\delta_{level}$ indicates elevated induction levels and a negative value of $\delta_{level}$ indicates lowered induction levels. To study how each parameter can affect the bimodal range, we next perturbed free parameters in our model to varying degrees and summarized how parameter perturbations affected bimodal ranges and induction levels in a matrix for subsequent analysis. We allowed 36 parameters to vary one at a time, and each multiplied by a perturbation factor from one of 0.01, 0.1, 0.5, 2, 10, and 100, thus the total parameter perturbations: $n = 36*6 = 216$. For each parameter condition, we fixed the glucose concentration and simulated titrating galactose from low to high or high to low concentrations to determine the bimodal region. This was repeated for eight glucose concentrations (including zero glucose).

## Supporting information

**S1 Fig. The simulated induction levels in comparison with the FACS data of S288C strain.** (a)—(c) The 96 nutrient conditions were split into 8 rows, each displaying galactose titration at a fixed glucose concentration. The dots represent experimentally measured induction levels of the GAL pathway and the lines represent simulated induction levels. Yellow color codes for ON peak positions (for experimental data) or the higher levels at steady state (for simulated results), and black color codes for OFF peak positions (for experimental data) or the lower levels at steady state (for simulated results). Color shades represent different modalities determined experimentally. (a) The comparison of wildtype S288C strain. An example of YFP signal histogram is shown on the right. (b) The comparison of S288C$^{mig1\Delta}$ strain. (c) The comparison of S288C$^{gal80\Delta}$ strain. (EPS)

**S2 Fig. Clustering parameter perturbations based on their effect on bimodal range and induction level.** Each row represents a parameter perturbation, each column represents a metric: $\delta_{ON}$, $\delta_{full}$, and $\delta_{level}$ in 8 different glucose concentrations. $\delta_{ON}$ = -1 means that the galactose concentration required to induce the GAL pathway is halved, for instance, from 0.125% to 0.0625%, reflecting a more inducible system. Similarly, $\delta_{full}$ = 2 means that the galactose concentration required to induce the GAL pathway increases fourfold, say, from 0.0625% to 0.25%, reflecting a more repressed system. $\delta_{level}$ is normalized to induction levels simulated with original parameters (the 'best-ft' parameter set without any perturbation). Curly braces highlighted the four groups of parameter perturbations based on their corresponding metrics. (EPS)

**S3 Fig. Zooming in on group 'more inducible'.** Enlarge the group 'more inducible' in S2 **Fig** to show parameter perturbations that constitutes this group. (EPS)

**S4 Fig. Zooming in on group 'more repressed'.** Enlarge the group 'more repressed' in S2 **Fig** to show parameter perturbations that constitutes this group. (EPS)

**S5 Fig. Zooming in on group 'bimodality insensitive'.** Enlarge the group 'bimodality insensitive' in S2 **Fig** to show parameter perturbations that constitutes this group. (EPS)

**S6 Fig. The effect of parameter perturbations on each metric.** (a)—(c) The perturbation effect on $\delta_{ON}$, $\delta_{full}$, and $\delta_{level}$, respectively. (EPS)

**S7 Fig. Parameter perturbations that make GAL network more inducible, more repressed, and bimodality insensitive.** (a)—(c) Each subplot shows the induction levels under a fixed glucose concentration. (a) The effect of perturbating kf83 on GAL induction profiles. When increasing kf83 by 2-fold, 10-fold, and 100-fold (reddish lines in the plot), a subpopulation of yeast cells show induction at lower galactose concentration, compared to the phenotype captured by default parameters (shown in cyan line). Increasing kf83 is an example of parameter perturbations that make GAL network more inducible. (b) The effect of perturbating ag80 on GAL induction profiles. While decreasing ag80 doesn't show pronounced effect on GAL induction (greenish lines in the plot, overlapped with the cyan line denoting default parameters), increasing ag80 substantially repressed GAL induction (reddish lines in the plot) even at high galactose concentration. Increasing ag80 is an example of parameter perturbations that make GAL network more repressed. (c) The effect of perturbating KR1 on GAL induction profiles. Either increasing KR1 (reddish lines in the plot) or decreasing KR1 (greenish lines in the plot) doesn't affect bimodal range, compared to the phenotype captured by default parameters (shown in cyan line). Tuning KR1 is an example of parameter perturbations that are bimodality insensitive. (EPS)

**S8 Fig. Examples of fitting I14 and YPS163 strains by tuning single parameter.** (a)—(d) Each subplot shows the induction levels under a fixed glucose concentration. Red circles denote the peak induction levels of ON cells in experimental data, black circles denote the peak induction levels of OFF cells in experimental data. Red lines denote the simulated highly induced state, black lines denote the simulated lowly induced state. (a) Fitting I14 strain by tuning rHXT, fitting improvement = 0.9, as shown in Fig 4. The tuned parameters capture the induced subpopulation (ON cells) at low galactose concentrations. In contrast, (b) fitting I14

strain by tuning KRs (fitting improvement = 0.6) or (c) fitting I14 strain by tuning KG1 (fitting improvement = 0.6), is not able to capture these early induced ON cells. (d) Fitting YPS163 strain by tuning KR1 (fitting improvement = 0.6).
(EPS)

**S9 Fig. Examples of fitting YPS606 and YPS163 strains by tuning single parameter.** (a)—(d) Each subplot shows the induction levels under a fixed glucose concentration. Red circles denote the peak induction levels of ON cells in experimental data, black circles denote the peak induction levels of OFF cells in experimental data. Red lines denote the simulated highly induced state, black lines denote the simulated lowly induced state. (a) Fitting YPS606 strain by tuning rG2, fitting improvement = 0.6, as shown in Fig 4. (b) Fitting YPS606 strain by tuning ag2 (fitting improvement = 0.7). (c) Fitting YPS606 strain by tuning kf3 (fitting improvement = 0.7). (d) Fitting YPS163 strain by tuning aR (fitting improvement = 0.7).
(EPS)

**S10 Fig. Examples of fitting 273614N and CLIB324 strains by tuning single parameter.** (a)—(d) Each subplot shows the induction levels under a fixed glucose concentration. Red circles denote the peak induction levels of ON cells in experimental data, black circles denote the peak induction levels of OFF cells in experimental data. Red lines denote the simulated highly induced state, black lines denote the simulated lowly induced state. (a) Fitting 273614N strain by tuning KG2, fitting improvement = 0.7, as shown in Fig 4. (b) Fitting 273614N strain by tuning a2 (fitting improvement = 0.7). (c) Fitting 273614N strain by tuning kf83 (fitting improvement = 0.8). (d) Fitting CLIB324 strain by tuning a1 (fitting improvement = 0.6).
(EPS)

**S1 Table. List of strains used in this study.**
(DOCX)

**S2 Table. Parameter descriptions, units, and 'best-fit' values.**
(DOCX)

## Acknowledgments

We thank Kayla Lee for sharing GAL induction profiles, Yihan Lin, Yanjun Li, and Ang Li for helpful discussions and comments, and Becky Ward for critically reading the manuscript.

## Author Contributions

**Conceptualization:** Jiayin Hong, Bo Hua, Michael Springer.

**Formal analysis:** Jiayin Hong.

**Funding acquisition:** Jiayin Hong, Michael Springer.

**Investigation:** Jiayin Hong, Bo Hua.

**Supervision:** Michael Springer.

**Validation:** Jiayin Hong, Julius Palme.

**Visualization:** Jiayin Hong.

**Writing – original draft:** Jiayin Hong.

**Writing – review & editing:** Jiayin Hong, Michael Springer.

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
