## [Decision Letter · Decision Letter 0]

22 Mar 2021

Dear Dr. Springer,

Thank you very much for submitting your manuscript "Computational analysis of GAL pathway pinpoints mechanisms underlying natural variation" for consideration at PLOS Computational Biology.

As with all papers reviewed by the journal, your manuscript was reviewed by members of the editorial board and by several independent reviewers. In light of the reviews (below this email), we would like to invite the resubmission of a significantly-revised version that takes into account the reviewers' comments.

We cannot make any decision about publication until we have seen the revised manuscript and your response to the reviewers' comments. Your revised manuscript is also likely to be sent to reviewers for further evaluation.

Sincerely,

Sergei L. Kosakovsky Pond, PhD

Associate Editor

PLOS Computational Biology

Douglas Lauffenburger

Deputy Editor

PLOS Computational Biology

Reviewer's Responses to Questions

**Comments to the Authors:**

Reviewer #1: Summary and relevance

Many genetic studies focus on unraveling the different alleles that contribute to a quantitative trait. Such QTL or GWAS studies typically only identify a specific genomic region, and thus require more experimental work to pinpoint the exact alleles that dive the observed phenotypic variation. Moreover, in many cases, the underlying molecular mechanism (i.e. how a given genetic variation might influence a quantitative trait) remains obscure. This study by Hong and coworkers shows how mathematical models can help to predict the possible molecular mechanisms underlying natural variation in a nutrient-sensing network. Such models could be a formidable tool to help predict and understand if and how specific genetic variation (alleles) might influence complex, quantitative phenotypes.

Specifically, the authors describe the development of a model that recapitulates the complex behavior of the galactose network in the eukaryotic model S. cerevisiae. The authors previously used a clever setup to characterize the dynamic, quantitative behavior (input-output response) of the GAL network and found that in general, yeast cells activate the GAL genes when the ratio of the concentration of galactose to glucose reaches a certain threshold. Around the threshold, the system is bimodal and unstable, resulting in variability within an isogenic population. Moreover, the exact sugar ratio required to activate or inactivate the network, as well as the bimodality and induction levels vary between different S. cerevisiae genetic backgrounds, making the GAL response a good model for a complex, quantitative trait. In this new work, using a model that recapitulates the GAL network, the authors show that the variation between the GAL response of natural yeast strains can be explained by changes in a few relatively simple parameters, such as the ration of the glucose and galactose binding affinity for the shared transporter Hxt, the binding rate between Gal3 and Gal80, the binding rate between galactose and Gal80 or the synthesis rate of Mig1. Hence, the model predicts that mutations/alleles that would affect these parameters could explain the observed natural variation in the GAL phenotypes.

general assessment:

Overall, the work is interesting to the broad range of colleagues interested in QTL analyses as well as those modeling regulatory pathways. The text is exceptionally well-written and the work seems solid. As such, I think it is perfectly suited for PLoS Comp Biol.

Major comment

My only concern is that the authors fail to validate their predictions. Ideally, the authors would complement this study with a QTL analysis (eg pooled segregant analysis) to pinpoint some of the alleles underlying the observed variation in some of the natural yeast strains, and then relate this back to their model, to show that the model indeed recapitulated reality. I realize that this is a lot to ask and goes beyond the scope of the paper. So instead, it might be possible to come up with some more simple validation tests, eg by changing Mig1 expression or mutating Gal80, investigate the effect of these genetic perturbations on the GAL behavior and discuss whether this is in line with what would be expected from the model? Or perhaps the authors have another idea how they could validate their predictions?

Minor:

Line 102 could use a reference.

Reviewer #2: In this manuscript, Hong et al. use an ODE model of the GAL pathway in yeast to try to explain variation in bimodality in among natural isolates. The Springer lab has been using the GAL pathway to explore the mechanistic basis of quantitative genetic variation, and this paper extends that work. They examined 15 natural isolates and found variation in the ratio of glucose to galactose at which the GAL pathway is induced as well as in the induction levels of the pathway. They built a somewhat simplified ODE model of the pathway (similar to other models in the literature) and fit it to match the induction behavior of a standard lab strain and two mutants with deletions of key genes in the pathway. Starting from this parameter fit, they systematically perturbed parameters to see if they could identify ones that affected the bimodal range and then to find ones which, when perturbed, produced a better match to the experimental data for the natural isolates.

In general this study, as presented, follows a logical progression. Natural variation is observed; a generative model is produced and fit to a reference strain(s); parameters of the model are perturbed to see if they can explain the natural variation.

The experimental data seems solid. The Springer lab has published these kinds of experiments for a while now and the new thing here is that they use new strains.

The model is a variant of commonly used ODE models of the GAL pathway. They include a few more details than other models about sugar uptake and intracellular sugar dynamics, which are useful additions, especially given the imporance of glucose:galactose ratios in inducing the pathway. One thing that isn't clear to me is whether the results shown in figure 2b are good fits to the experimental data. There are substantial quantitative differences in the GAL induction levels for all three strains between the simulations and experiments. This is not terrible - the model is a simplification after all - but the authors should discuss it and discuss the implications of this less than ideal fit for the investigations that follow. They used an objective function that summarized the differences between this heat maps but there isn't any insight given into how far off the simulations were and what the landscape looks like. It's also important to note that for some of the parameters (and some key parameters like rHXT and ag2) the wildtype parmeter distribution spanned at least 1 and sometimes close to 2 orders of magnitude. Later, when they explore the effect of parameter value perturbations, they do it over 4 orders of magnitude (2 up, 2 down). Does it matter that these particular parameters are not very well constrained?

Figures S2-S5 show the effects of changing various parameters on the deltaON, deltaFULL, and deltaLEVEL metrics. These figures are hard to read. There is a lot going on, and I'm not sure how to fix them. Perhaps splitting the ON,FULL,lEVEL into their own groups instead of having them side by side within each glucose concentration might help.

In Figure 4, they show how changing various parameters improves the fit of the model. This is based on equation 2. What does "min" in this equation mean? What is the minimization taken over? On that note, for these natural isolate fits, are the fits just tested at the 2-fold, 10-fold, 100-fold levels or are they allowed to slide continuously, one parameter at a time?

In line 292, the authors say they simulated the induction profile for 8 strains. How does this differ from the assessment they did in the previous paragraph? Didn't they already have the simulation results? Why would they need to simulate them again? If I am misunderstanding the steps and the difference between the two parts, please make them a little more clear.

As with figure 2, for figure 5 I can look at the simulated vs. experimental heat maps and accept that they look similar in some ways and fir some strains more than others, but there are clear differences and perhaps changing a different parameter would also make a heat map that looked acceptably close. It would be nice to see a graph with a quantification of how much better changing this parameter did than changing the others and how close it got to the actual data.

This leads to my final point. The authors have used the model to make a clear hypothesis for what is causing the differences between the strains, or at least the ones in figure 5. But at this point, it's just a hypothesis, and we don't really know whether this whole exercise was useful in explaining the actual difference between natural isolates or just an exercise in exploring the behavior of the model. It would be nice to have some experimental verification. This isn't trivial, especially for the rHXT parameter since this is a composite of several genes, but perhaps YJM981 and YPS606 could be tested with allele swaps. For YPS606 they did this allele swap in a previous paper (at least in one direction). It isn't clear to me whether they did the full heat map, but they presumably already have the strain. For YJM981, perhaps a MIG1 promoter swap would be appropriate. It would be even better if they could figure out a way to swap the Gal80-Gal3 binidng interfaces so as to affect the kf83 parameter because then they could test three of their strains. Again, it isn't necessarily straightforward to do these (except perhaps for YPS606), but it would provide an unassailable validation of their approach.

Minor note: line 678 - "Besides" is the wrong word.

line 368 - "simply" is the wrong word.

Reviewer #3: In this work, the authors developed an ODE model of the essential parts of the GAL metabolic network in s. cerevisiae that captures the experimental response of natural isolates of yeasts to combinations of glucose and galactose.

The authors performed fit of the model to WT, gal80d, and mig1d and show that, besides the parameters that are associated with the delete, the same parameters capture induction levels of all strains.

The authors then perform a parameters scan and quantify the changes in three parameters that are relevant to the bimodal nature of the system and identify a few typical classes of perturbation. The authors also quantify the difference in the parameters the fit the data of the different natural strains. Finally, the author demonstrates how a change of single parameters can capture the changes in the decision fronts.

The goal to connect macroscopic phenotypic changes to changes in microscopic parameters is important and interesting. The GAL system provides a simple yet rich system to demonstrate the value of connecting microscopic models to macroscopic phenomena.

I have a few minor comments:

1) Decision front: It would be beneficial to explain more about why the decision fronts are straight lines (in the log-log space) and more details on how they are calculated (half maximum of the whole plate? of isoglucose line?) and why they span differently in different strains.

2) Add a plot of the phase space that will allow estimating the hysteresis of the system.

3) When analyzing the bimodality, it would be interesting to add the natural strain distribution on the tSNE to see the natural variation span with respect to the clusters. Along the same lines, it would be insightful to perform some non-linear embedding analysis (e.g., diffusion map) to the natural stains data.

4) It would be insightful to include a figure that compares the region of bimodality with the best fit.

5) Including representative examples for the actual dynamics of each cluster of 3b would help to understand the change at each cluster.

6) what is the impact of the single parameter changes that explain the decision front movement (fig. 5) on the bimodality parameters? is the change consistent?

7) The first sentences in the abstract that mention human traits seem detached from the rest of the paper and should be removed.

**Have all data underlying the figures and results presented in the manuscript been provided?**

Reviewer #1: Yes

Reviewer #2: None

Reviewer #3: None

PLOS authors have the option to publish the peer review history of their article (what does this mean?). If published, this will include your full peer review and any attached files.

Reviewer #1: **Yes: **Kevin J. Verstrepen

Reviewer #2: No

Reviewer #3: No
---

## [Decision Letter · Decision Letter 1]

17 Aug 2021

Dear Dr. Springer,

We are pleased to inform you that your manuscript 'Computational analysis of GAL pathway pinpoints mechanisms underlying natural variation' has been provisionally accepted for publication in PLOS Computational Biology.

Before uploading files to production, please make an effort to update the figures as requested by Rev 2.

Best regards,

Sergei L. Kosakovsky Pond, PhD

Associate Editor

PLOS Computational Biology

Douglas Lauffenburger

Deputy Editor

PLOS Computational Biology

Reviewer's Responses to Questions

**Comments to the Authors:**

Reviewer #1: I congratulate the autthors with their interesting study and thank them for responding to my concerns; I feel that the addition of the new data to help validate the predictions make the paper even stronger and as far as I am concerned, it is ready for publication.

kevin verstrepen

Reviewer #2: This is a revision of a previously submitted manuscript. The authors have carefully gone through the comments and addressed them for the most part. In my opinion, however, they seem to fall on the side of presenting too little information rather than too much in their supplementary figures. For example, in figure S1, they show how far off the simulations are from experimental data - but only for wild-type, and they say that the fit was not as good for mig1_delta and gal80_delta. I would encourage them to show all three. Similarly to go into a little more depth in figure 4, they have prepared figure S8 which is an interesting figure and gives a lot more insight into what rHXT and KRs are doing than the two boxes in figure 4. As the authors themselves suggest, the fit can be improved in different ways. I assume the authors have these curves for all their strains x parameters. They could include more of them, say some subset of the squares in figure 4 above 0.5. This would be a big figure, but it in the supplementary information that shouldn't be a big deal and would be of great interest to a key part of the audience for this paper.

Minor comments: Line 482. Is this 6 orders of magnitude? Isn't it 4? 1/100 <-> 1/10 <-> 1 <-> 10 <-> 100?

Figs S7,S8. There are two y-axes - one for the overall grid and one for each individual plot. The individual ones are not labeled. There isn't much space in the figure but if nowhere else then at least in the caption the authors should say explicitly that this is induction level.

Overall, it is a nice paper and a worthy addition to this line of research from the Springer lab.

Reviewer #3: The authors have responded in a satisfying manner to all my comments.

**Have the authors made all data and (if applicable) computational code underlying the findings in their manuscript fully available?**

Reviewer #1: Yes

Reviewer #2: Yes

Reviewer #3: Yes

PLOS authors have the option to publish the peer review history of their article (what does this mean?). If published, this will include your full peer review and any attached files.

Reviewer #1: **Yes: **Kevin J. Verstrepen

Reviewer #2: No

Reviewer #3: No

---

## [Editor Report · Acceptance letter]

21 Sep 2021

PCOMPBIOL-D-21-00037R1 

Computational analysis of GAL pathway pinpoints mechanisms underlying natural variation

Dear Dr Springer,

I am pleased to inform you that your manuscript has been formally accepted for publication in PLOS Computational Biology. Your manuscript is now with our production department and you will be notified of the publication date in due course.

With kind regards,

Amy Kiss
